# NLRP3 Inflammasome Involved with Viral Replication in Cytopathic NADL BVDV Infection and IFI16 Inflammasome Connected with IL-1β Release in Non-Cytopathic NY-1 BVDV Infection in Bovine Macrophages

**DOI:** 10.3390/v15071494

**Published:** 2023-06-30

**Authors:** Claudia Gallegos-Rodarte, Omar Escobar-Chavarría, Meztli Miroslava Cantera-Bravo, Rosa Elena Sarmiento-Silva, Alejandro Benitez-Guzman

**Affiliations:** Departamento de Microbiología e Inmunología, Facultad de Medicina Veterinaria y Zootecnia, Universidad Nacional Autónoma de México, Mexico City 04510, Mexico; vetrodarte@gmail.com (C.G.-R.); omare@fmvz.unam.mx (O.E.-C.); 16mc05meztlicantera@gmail.com (M.M.C.-B.); rosass@unam.mx (R.E.S.-S.)

**Keywords:** inflammasome, viral replication, caspase-1, BVDV

## Abstract

Inflammasomes are multiprotein complexes that play a role in the processing of proinflammatory cytokines such as interleukin 1 beta (IL-1β). The secretion of IL-1β in bovine macrophages infected with the bovine viral diarrhea virus (BVDV) cytopathic strain NADL (NADLcp-BVDV) is caspase 1-dependent. In the present study, we found that in macrophages infected with NADL, the NLRP3 inflammasome participated in the maturation of IL-1β as the level decreased from 4629.3 pg/mL to 897.0 pg/mL after treatment with cytokine release inhibitory drug 3 (CRID3). Furthermore, NLRP3 activation has implications regarding viral replication, as there was a decrease in the viral titer until 1 log of a supernatant of macrophages that were inhibited with CRID3 remained. In the case of the non-cytopathic BVDV strain NY-1 (NY-1 ncpBVDV), IL-1β secretion is not affected by NLRP3, but could be related to the IFI16 inflammasome; we found a colocalization of IFI16 with ASC using confocal microscopy in infected macrophages with the NY-1 ncp-BVDV biotype. To relate IFI16 activation to IL-1β release, we used ODN TTAGGG (A151), a competitive inhibitor of IFI16; the results show a decrease in its level from 248 pg/mL to 128.3 pg/mL. Additionally, we evaluated the caspase 1 activation downstream of IFI16 and found a decrease in the IL-1β from 252.9 pg/mL to 63.5 pg/mL when caspase 1 was inhibited with Y-VAD. Our results provide an improved understanding of the mechanisms involved in the viral replication, inflammation and pathogenesis of bovine viral diarrhea.

## 1. Introduction

Bovine viral diarrhea, caused by the bovine viral diarrhea virus (BVDV), can be found worldwide and can cause significant economic losses due to its diverse clinical manifestations. The severity of this disease depends on the characteristics of infected individuals, such as gestational stage and immune status, as well as the viral genotype and strain [1]. BVDV strains are classified into two genotypes and two biotypes, depending on the effects they produce in in vitro-cultured cells. The cytopathic biotype (cp) leads to cell death, while the non-cytopathic biotype (ncp) has no apparent effects in cultured cells and is related to viral persistence in cattle. The BVDV belongs to the genus *Pestivirus* of the family *Flaviviridae*. This viral family contains several viruses that can be detrimental to human health, such as the hepatitis C, dengue, Zika, yellow fever and Weast Nile viruses, among others [2,3].

Macrophages are susceptible to the BVDV due to the tropism of the virus toward cells of the immune system, leading to lymphopenia, leukopenia and thrombocytopenia in infected animals [4,5,6,7]. Once the virus infects macrophages via interaction with the CD46 receptor or the LDLr (low-density lipoprotein receptor), they can be internalized by clathrin-mediated endocytosis [8,9,10]. The macrophages possess pattern recognition receptors (PRRs) such as Toll-like receptors (TLRs), nucleotide oligomerization domain-like (NOD-like) receptors and retinoic-acid-inducible gene-like (RIG-like) receptors, which can recognize viral components, triggering signal transduction pathways to produce diverse cytokines [11]. Among these receptors is NLRP3, an NOD-like receptor that, when activated, can form multiprotein complexes known as inflammasomes. The activation of NLRP3 through recognition of pathogen-associated molecular patterns (PAMPs) or damage-associated molecular patterns (DAMPs) prompts structural changes that allow the interaction of the pyrin domain (PYD) of NLRP3 with the adapter protein ASC (apoptosis-associated speck-like protein) through homotypic interactions. After this, ASC can interact, via its CARD (caspase activation and recruitment domain), with pro-caspase 1 to give rise to active caspase 1, which can perform proteolytic processing to secrete IL-1β in its active form [12,13]. Other inflammasomes that have been reported during viral infections are AIM2 (absent in melanoma 2) and IFI16 (interferon-gamma-inducible protein 16). Both are ALRs (AIM-like receptors), and their activation is mediated by DNA or RNA; once activated, they interact downstream with ASC, inducing the recruitment of caspase 1 [14,15]. Viral replication causes a variety of cellular alterations that can stimulate the activation of inflammasomes such as viral protein aggregates and endosome rupture as well as the presence of reactive oxygen species (ROSs), the release of ATP, ion flow imbalance and direct or indirect interaction with viral proteins during viral replication [16,17,18,19,20,21].

In BVDV infection, differential expressions of proinflammatory cytokines related to the biotype, such as pro-IL-1β, have been reported. It has been demonstrated that the expression of pro-IL-1β is higher with highly virulent non-cytopathic biotypes than with low-virulence non-cytopathic biotypes [22,23]. However, the protein secretion of IL-1β during infection with non-cytopathic biotype strains is lower than that with cytopathic strains; this could be related to the processing of IL-1β by inflammasomes [24,25]. Differences in gene expression have been observed between infection with cytopathic and non-cytopathic biotypes, with increased gene expression of NFкB shown during infection with the cytopathic biotype and an increase in IFN-stimulated genes observed during infection with the non-cytopathic biotype [26,27].

In a previous study, our research group reported that the secretion of IL-1β was caspase 1-dependent during infection of bovine macrophages with the NADL cp-BVDV strain [25]. Furthermore, we observed a decrease in the viral titer of a supernatant of macrophages infected with the NADL cp-BVDV strain when they were treated with a caspase 1 inhibitor (Y-VAD, N-acetyl-tyrosyl-valyl-alanyl-aspartyl chloromethylketone), suggesting that this protease may be related to viral replication. However, that study did not elucidate the activation of the inflammasome upstream of caspase 1 or its relationship to IL-1β release and replication. Therefore, in the present study, we evaluated this activation during infection of bovine macrophages with the NADL cp-BVDV and NY-1 ncp-BVDV strains.

## 2. Materials and Methods

### 2.1. Ethical Approval

All animal-related procedures in this study were carried out in accordance with the guidelines established by the Institutional Animal Care and Use Committee (SICUAE) of the Faculty of Veterinary Medicine and Zootechnics of the National Autonomous University of Mexico (SICUAE.MC-2020/3-1).

### 2.2. Macrophage Cultures

Blood samples from adult cattle were obtained from the Center for Practical Teaching and Research on Animal Production and Health (Centro de Enseñanza Práctica e Investigación en Producción y Salud Animal; CEPIPSA) of the National Autonomous University of Mexico. Macrophages were obtained from mononuclear cells of peripheral blood using the method previously described by Morales et al., 2020 [25]. Briefly, blood samples were taken using jugular venipuncture with 60 mL syringes of ACD (acid citrate dextrose) anticoagulant. The whole-blood samples were transferred to polypropylene tubes and centrifuged at 1200× *g* for 30 min to obtain a buffy coat, which was submitted to a density gradient using Histopaque^®^-1077 (Sigma-Aldrich, Saint Louis, MO, USA) to obtain a layer that contained monocytes. The monocytes were washed with PBS–citrate and cultured on ultra-low-adherence plates with an RPMI medium supplemented with 12% autologous serum. The medium was changed every three days until use 12 days later.

### 2.3. Viral Propagation and Titer

To propagate and acquire the viral stocks of the NADL cp-BVDV and NY-1 ncp-BVDV strains, we used Madin–Darby Bovine Kidney Cells (MDBK) (ATCC^®^ CCL-22, Manassas, VA, USA). We performed RT-PCR to confirm the absence of the BVDV in the MDBK cells and fetal bovine serum, using the primers FW GCTAGCCATGCCCTTAGTAGGACTAGC and RV AACTCCATGTGCCATGTACAGCAGAG to amplify the 5′ UTR region. The fetal bovine serum was free of BVDV antibodies. The cells were maintained in a DMEM supplemented with 10% fetal bovine serum (FBS) (Gibco, New York, NY, USA) and incubated at 37 °C with 5% CO_2_. To infect the cells, we changed the medium to a DMEM supplemented with 2% FBS, agitated it for 2 h and then incubated it for 72 h for the NADL cp-BVDV strain and 96 h for the NY-1 ncp-BVDV strain; the cells were lysed through freeze–thawing (3 cycles) to obtain the viral stock for each strain. For the viral titration of the NADL cp-BVDV strain, we cultured MDBK cells on 96-well plates, performing tenfold serial dilutions of the viral inoculum, and the titer was determined with the Reed–Müench Method. For the viral titration of the NY-1 ncp-BVDV strain, we used BT cells (Bos Taurus turbinate, ATCC^®^ CRL-1390, Manassas, VA, USA) cultured with a DMEM supplemented with 10% horse serum on 24-well plates with 12 mm coverslips. The cells were infected with the viral strain, and at 72 h post-infection, they were removed from the medium and fixed on slides with PBS 1X-PFA 4% for 15 h at 4 °C and washed 3 times with PBS 1X. They were then incubated in a blocking buffer (PBS1X/5% serum/0.3% tritonX100) for 1 h at room temperature, after which the blocking buffer was removed and washed and the cells were incubated with anti-BVDV polyclonal antibodies (VMRD^®^ BVDV FITC, Washington, DC, USA) for 16 h at 4 °C. Then, the antibodies were removed, a Vectashield^®^ mounting medium with DAPI (Vector Laboratories, Newark, CA, USA) was added, and the fluorescent foci were counted using a fluorescence microscope (Leica DM1000, Leica, Wetzlar, Germany) to determine the focus-forming units per mL (FFUs/mL).

### 2.4. Macrophage Infection

The bovine macrophages were cultured on 24-well plates, at a density of 3 × 10^5^ cells per well, to perform infection assays. We used multiplicities of infection (MOIs) of 2:1 and of 10:1 for the NY-1 ncp-BVDV strain and 2:1 for NADL cp-BVDV strain, with 500µL of RPMI medium supplemented with 10% heat-inactivated FBS (Gibco, New York, NY, USA). The negative controls were macrophages cultured with RPMI and 10% FBS, and the positive control was treated with 300 ng/mL of LPS from *E. coli* type O26:B26. For the inhibition treatments, the macrophages were pre-incubated for 2 h with CRID3 (Cytokine Release Inhibitory Drug 3, 50 µM; Sigma Aldrich, St. Louis, MO, USA), Y-VAD (50 µM; Merck, Boston, MA, USA) or Z-VAD (carbobenzoxy-valyl-alanyl-aspartyl-[O-methyl]- fluoromethylketone, 50 µM; Sigma Aldrich, St. Louis, MO, USA) and ODN A151 (5′-TTAGGGTTAGGGTTAGGGTTAGGG-3′) (3 µM, 6 µM and 12 µM). The supernatants were recovered at 24 h post-infection.

### 2.5. Quantification of IL-1β

We quantified IL-1β secretion in the supernatants from the infection assays using the IL-1β Bovine Uncoated ELISA Kit (Thermo Fisher, Wien, Austria). The plates were read in a spectrophotometer at a wavelength of 450 nm. The results were analyzed with GraphPad Prism 8^®^ software for interpretation.

### 2.6. Quantification of ASC Specks

Bovine macrophages (1 × 10^4^) on cell culture slides were infected with either the NADL cp-BVDV or NY-1 ncp-BVDV strain (MOI 2:1), either with or without pre-incubation with the NLRP3 inhibitor CRID3 2 h prior to infection. Then, the cells were washed with PBS and fixed with paraformaldehyde (PFA) at (pH 7.4) for 10 min, permeabilized with triton 100× at 0.15% for 10 min on ice and washed with PBS and incubated overnight with anti-TMS1/ASC antibodies (Abcam, Cambridge, UK) at a dilution of 1:1000 and at 4°. The cells were washed and incubated with donkey anti-goat IgG HL (FITC) secondary antibodies (Abcam, Cambridge, UK) at 1:3000 for 1 h while being protected from light. The slides were washed and mounted using a Vectashield^®^ medium with DAPI (Vector Laboratories, CA, USA). Images thereof were acquired with a fluorescence microscope (Leica DM1000, Leica, Wetzlar, Germany). The same capture parameters, exposure time and stain intensity were applied for the control and experimental samples.

Quantification of the ASC specks was performed by taking the average of the positive cells (cells with ASC specks) out of the total cells per field. We counted 30 fields for each treatment, and the procedure was performed in triplicate, using the total count for all repetitions to perform the statistical analysis.

### 2.7. Confocal Microscopy

Bovine macrophages (1 × 10^4^) on cell culture slides were infected with NY-1 ncp-BVDV strain MOI 2:1 and BoHV-1 (bovine herpesvirus-1) MOI 10:1 as a positive control; the negative control consisted of macrophages with RPMI 10% FBS. The cells were washed with PBS and fixed with PFA at 4% in PBS (pH 7.4) for 10 min at 24 h post-infection. Then, they were incubated in a blocking buffer (PBS 1X/5% serum/0.3% triton X100) for 1 h at room temperature. After that, the blocking buffer was removed and washed; then, the cells were incubated overnight with primary anti-TMS1/ASC antibodies (Ab155449 Abcam, Cambridge, UK) and primary anti-IFI16 antibodies (D8B5T Cell Signaling, Danvers, MA, USA) at a dilution of 1:500. Following the incubation, the cells were washed with 1X PBS–0.05% Tween 20 and incubated with secondary donkey anti-goat IgG antibodies (Alexa Fluor 594, Abcam, UK) in a dilution buffer (1X PBS/1%BSA/0.3% Triton X100); after a 1 h incubation period at room temperature, the cells were washed with 1X PBS–0.05% Tween 20 again and then incubated with secondary goat anti-rabbit IgG antibodies (Alexa Fluor 488) for 1 h at room temperature. The slides were washed and mounted using a Vectashield^®^ medium with DAPI (Vector Laboratories, CA, USA). Images thereof were acquired with a confocal microscope (Nikon A1R + STORM, Tokyo, Japan).

### 2.8. MTT Assay

Bovine macrophages were cultured in 24-well plates at a density of 3 × 10^5^ cells per well. The cells were incubated for 24 h with 3 μM, 6 μM and 12 μM of an A151 inhibitor. After incubation, the cells were cultured with RPMI 10% FBS and MTT (3-(4,5-dimethylthiazol-2-yl)-2,5-diphenyltetrazolium bromide) for 4 h. The positive control, based on H_2_O_2_, was incubated for 6 h, and the negative control was composed of cells with RPMI 10% FBS. An MTT assay was performed based on the manufacturer’s specifications (Roche, Cell Proliferation Kit I; Basilea, Switzerland). The supernatants were measured in a 96-well plate and read with a spectrophotometer at a wavelength of 570 nm.

### 2.9. Statistical Analysis

Data are expressed as means ± standard deviations. The significances of differences between any two groups were evaluated using Student’s *t*-test, and differences among three or more groups were evaluated using one-way ANOVA. *p*-values < 0.05 were considered significant. All analyses were carried out using GraphPad Prism version 8 software for Windows (GraphPad Software, San Diego, CA, USA).

## 3. Results

### 3.1. IL-1β, Secreted by Bovine Macrophages, Infected with the NADL cp-BVDV and NY-1 ncp-BVDV Strains

It has been reported that IL-1β production and inflammation can activate the replication of some viruses. For this work, we wanted to know if infection with cytopathic NADL and non-cytopathic NY-1 strains of BVDV would promote expression of IL-1β. To solve this question, we quantified the secretion of IL-1β by macrophages infected with the BVDV. We observed that the NADL cp-BVDV-infected macrophages expressed 4629.3 pg/mL of IL-1β (Figure 1A). In the case of the NY-1 ncp-BVDV strain, to determine the effect of infection on IL-1β secretion, we used MOI 2:1 and 10:1 for 24 h, obtaining secretion levels of 252.9 pg/mL and 306.5 pg/mL, respectively (Figure 1B), with no difference between the MOIs. These results show that the secretion of IL-1β was up to 18-fold higher in the cytopathic biotype than in the non-cytopathic biotype.

### 3.2. IL-1β Secretion Decreased after Treatment with Caspase Inhibitors Y-VAD and Z-VAD

Treatment with the caspase 1-specific inhibitor Y-VAD decreased IL-1β secretion for both strains from 4629.3 pg/mL (without Y-VAD) to 1294.3 pg/mL (with Y-VAD; Figure 2A) during the infection with the NADL cp strain and from 252.9 pg/mL (without Y-VAD) to 63.5 pg/mL (with Y-VAD) with the NY-1 ncp strain (Figure 2B). These results suggest that for both strains, caspase 1 is involved in the processing of IL-1β. This is similar to previous reports for the cytopathic biotype and is a novel result for the non-cytopathic biotype. When we used the pancaspase inhibitor Z-VAD, IL-1β secretion was lower (550.1 pg/mL) during the infection with the NADL cp strain, which could suggest the involvement of other caspases in the secretion of IL-1β; in the case of the NY-1 ncp strain, there was no statistical significance (Figure 2B).

### 3.3. NLRP3 Inhibitor CRID3 Decreases IL-1β Secretion during NADL cp-BVDV Infection

To determine whether the NLRP3 inflammasome participates in the secretion of IL-1β during infection with the BVDV, we used the inhibitor CRID3 (50 µM), which interferes with the activation of NLRP3 upstream from caspase 1. We observed a decrease in IL-1β secretion from 4629.3 pg/mL (without CRID3 treatment) to 897.1 pg/mL (with CRID3 treatment; Figure 3A) in the macrophages infected with the NADL cp strain. There was a similar effect in the positive-control LPS (300 ng/µL), in which IL-1β secretion decreased from 3863 pg/mL without CRID3 treatment to 686.2 pg/mL with CRID3 treatment (Figure 3A). In the macrophages infected with the NY-1 ncp strain, treatment with CRID3 did not result in a significant difference in IL-1β secretion (Figure 3B), which could suggest that NLRP3 does not participate in IL-1β secretion in this strain. These results could indicate the participation of another inflammasome or of non-canonical pathways for activation of caspase 1 during infection with this strain.

### 3.4. Adaptor Protein ASC Recruited by NLRP3 Inflammasome during Infection with BVDV

ASC is an adaptor protein that possesses a pyrin domain that interacts with NLRP3 once it is activated, resulting in the formation of aggregates for the activation of caspase 1. During infection with the NADL cp strain in bovine macrophages, those without CRID3 treatment showed ASC speck formation in 19.1% of the cells (Figure 4A,B). This decreased significantly among the cells treated with CRID3 (9.8%; Figure 4A,B), similarly to what occurred with the positive-control LPS (Figure 4A,B). This suggests that the recruitment of ASC during infection with the NADL cp strain in bovine macrophages is dependent on NLRP3. During infection with the NY-1 ncp strain, there was a lower speck formation percentage (10.3%) than with the NADL cp strain (Figure 4A,C), and the speck formation further decreased (5.3%) when the cells infected with the non-cytopathic biotype were pre-treated with CRID3 (Figure 4A,C).

### 3.5. The Relationship between the IFI16 Inflammasome and IL-1β Secretion during NY-1 ncp-BVDV Infection

We observed that caspase 1 activation was necessary for IL-1β release during NY-1 ncp-BVDV infection (Figure 2B); however, the treatment with CRID3 did not result in a significant difference in IL-1β secretion (Figure 3B). Because of this, our results suggest that other inflammasomes could participate in the secretion of IL-1β. In order to complete this assumption, we used confocal microscopy to observe the IFI16 and ASC colocalization in the cytosol of the macrophages infected with the NY-1 ncp-BVDV strain (Figure 5A). Afterward, to associate this interaction with the IL-1β release, we used A151 at different concentrations as a competitive inhibitor of IFI16 (3 µM, 6 µM and 12 µM). We observed a decrease in the IL-1β secretion dose response from 248 pg/mL without the inhibitor to 242.8 pg/mL with 3 µM, 177.1 pg/mL with 6 µM and 128.3 pg/mL with 12 µM A151 (Figure 5B). None of these concentrations induced cytotoxicity (Appendix A).

### 3.6. CRID3 Inhibitor Decreases Viral Replication of NADL cp-BVDV in Bovine Macrophages

To determine the effect of NLRP3 inhibition in viral replication, we determined the viral titer in supernatants obtained from the infection assays of the bovine macrophages. We observed the cytopathic effect (detachment of the monolayer and vacuolization) in the supernatant of the cells (Figure 6A) infected with the NADL cp strain and without CRID3 treatment until a dilution of 1/729 was reached, while for the supernatant of the cells treated with CRID3 (50 µM), the cytopathic effect was observed at a dilution of 1/81, resulting in viral titers of 10^4.23^ TCID_50_ (log_10_ 1.7 × 10^4^) and 10^3.32^ TCID_50_ (log_10_ 2.1 × 10^3^), respectively, with a decrease of one logarithm (Figure 6B). In the case of the NY-1 ncp strain, we found no effect of CRID3 on the viral titers of the supernatants of the infected macrophages (1.3 × 10^5^ FFUs/mL without CRID3, 1.5 × 10^5^ FFUs/mL with CRID3; Figure 6C). These results suggest that in the case of the NY-1 ncp strain, the inhibition of NLRP3 had an effect on the viral replication.

## 4. Discussion

Inflammasomes can be activated by diverse stimuli that are present during viral replication. The activation and inhibition of inflammasomes have been studied to determine their direct and indirect effects on viral replication, which can be applied to the development of possible therapeutic targets. The results of our study suggest that NLRP3 participates in NADL cp-BVDV infection of bovine macrophages; this was evidenced by a significant decrease in the IL-1β secretion in macrophages infected with NADL cp-BVDV when they were pre-incubated with the NLRP3 inhibitor CRID3. This is similar to previous findings during infections with other viruses of the family *Flaviviridae*, including the classical swine fever, hepatitis C, influenza A, Zika and dengue viruses, among others [28,29,30].

Previous studies involving cp-BVDV infection in macrophages have shown that IL-1β secretion is stimulated 24 h post-infection due to the activation of NFкB and that IL-1β maturation is mediated by the NLRP3 inflammasome [22,31]. Our viral titer assays suggest the possible participation of NLRP3 during replication of the NADL cp-BVDV strain, since there was a decrease in the viral titer of the supernatant of the macrophages treated with the CRID3 inhibitor prior to infection with the NADL cp-BVDV strain. These results are similar to those observed during infection of NLRP3-knockout THP-1 cells by the Zika virus, in which there was also a decreased viral titer related to the interaction of the viral protein NS1 and procaspase-1, as well as a decrease in IFN-1 [32]. A similar case occurred during an infection of CRIB cells with BoHV-1, which tested the activations of IFI16 and NLRP3 upstream of caspase 1 and found a decrease in the viral titer of cells treated with glyburide, which inhibits the indirect action of the inflammasome [33]. Our results suggest that during infection with the NADL cp-BVDV strain, inflammasome activation could be related with viral replication [34].

In the present study, our finding was novel for the NY-1 ncp-BVDV strain. IL-1β release was observed to be caspase 1-dependent, and upstream, it might be related with IFI16 activation during NY-1 ncp-BVDV infection in macrophages. This has been observed during other RNA viral infections, as was reported by Wichit et al., 2019 for Chikungunya and Zika virus infections in HFF1 and HEK293T cells with overexpression of IFI16, which had an antiviral effect associated with the type I IFN pathway [14]. Another recent study has shown IFI16 to have direct interaction with influenza A viral RNA in A549 cells as an upregulator of RIG-I [35]. In a study by La Polla et al., 2022 an increase in IFN-stimulated genes was shown with transcriptomic analysis in bovine lung primary cells (BPCs) infected with NY-1 ncp-BVDV [26]. This pathway is directly associated with IFI16 activation. Our results give a first approximation of IFI16 inflammasome activation during BVDV replication in bovine macrophages; however, more studies are needed to determine its activation mechanism during infection with the ncp-BVDV strain.

Furthermore, our results show decreased formation of ASC specks with CRID3 treatment during infection with the NY-1 ncp-BVDV strain. This could suggest the participation of some viral proteins with proteolytic functions that could lead to the maturation of caspase 1, as has been reported for other viral infections in which relationships between viral proteins and caspases and their involvement during replication have been observed. For example, during the replication of the hepatitis C virus (HCV) and the classical swine fever virus (CSFV), the participations of caspase 3 and caspase 6 have been shown to be involved in the processing of the NS5A protein [36,37,38]. Another example is the ORFV119 protein of the orf virus, which can activate caspase 8 and caspase 9, which are related with intrinsic and extrinsic apoptotic pathways [39]. Similarly, it is possible that the NY-1 ncp-BVDV strain could have a direct effect via the interaction of its proteins or an indirect effect via the activation of signaling pathways due to the activation of caspase 1 for later secretion of IL-1β. However, further studies on this biotype and other high- and low-virulence, non-cytopathic strains of the BVDV are needed in order to reach a better understanding of the mechanisms of virulence that could be participating [20,40].

In conclusion, our study results show the participation of the NLRP3 inflammasome in the secretion of IL-1β during infection with the NADL cp-BVDV strain; furthermore, we found a possible relationship between the activation of the NLRP3 inflammasome and viral replication. However, in NY-1 ncp-BVDV, the IL-1β release was NLRP3-independent but caspase 1-dependent. Our results show a possible relationship between the IFI16 inflammasome and IL-1β secretion during infection with this strain; nevertheless, it is important to elucidate the participation of this inflammasome during ncp-BVDV replication (Figure 7). These findings contribute to a better understanding of the immune response mechanisms that are involved in the recognition of cytopathic and non-cytopathic BVDV, as well as the viral mechanisms that are involved in the pathogenesis of this disease.

## Figures and Tables

**Figure 1 viruses-15-01494-f001:**
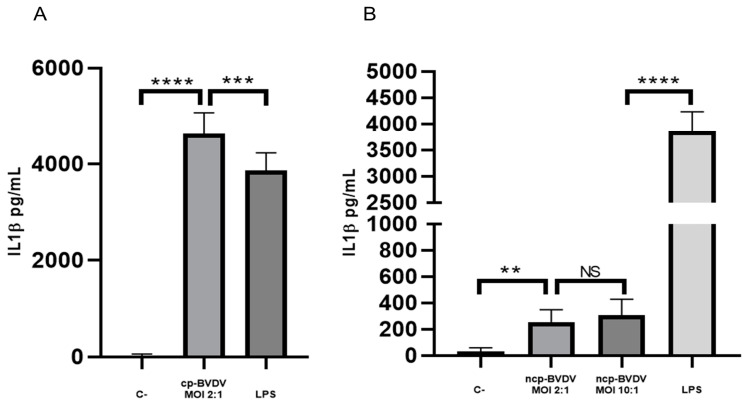
Secretion of IL-1β in bovine macrophages infected with NADL cp and NY-1 ncp strains. IL-1β secreted by the supernatants of macrophages infected with NADL cp-BVDV 24 h post-infection (MOI 2:1) (**A**) and NY-1 ncp-BVDV 24 h post-infection (MOI 2:1 and MOI 10:1) (**B**). Negative control (C-): supernatant of uninfected macrophages in RPMI medium with 10% FBS; LPS: supernatant of macrophages stimulated with LPS (300 ng/μL). Statistical differences are given as means ± SD values of three independent assays with two internal replicas each (**** *p* < 0.0001; *** *p* < 0.001, ** *p* < 0.01, NS: not significant *p* > 0.05).

**Figure 2 viruses-15-01494-f002:**
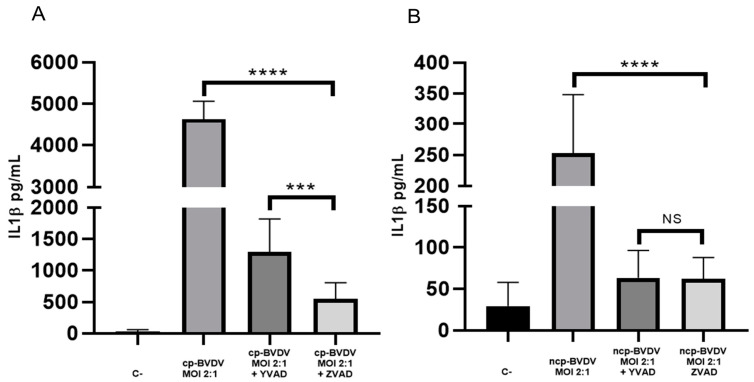
IL-1β secretion is caspase 1-dependent. Secretion of IL-1β in bovine macrophages infected with NADL cp and NY-1 ncp strains was determined with ELISA. IL-1β secreted by the supernatants of macrophages infected with NADL cp-BVDV strain 24 h post-infection (MOI 2:1) (**A**) and NY-1 ncp-BVDV strain 24 h post-infection (MOI 2:1) (**B**). Treatment with the inhibitors Y-VAD (50 μM, 2 h pre-incubation) and Z-VAD (50 μM, 2 h pre-incubation). Negative control (C-): supernatant of uninfected macrophages in RPMI medium with 10% FBS. Statistical differences are given as means ± SD values of three independent assays with two internal replicas each (**** *p* < 0.0001; *** *p* < 0.001, NS: not significant *p >* 0.05).

**Figure 3 viruses-15-01494-f003:**
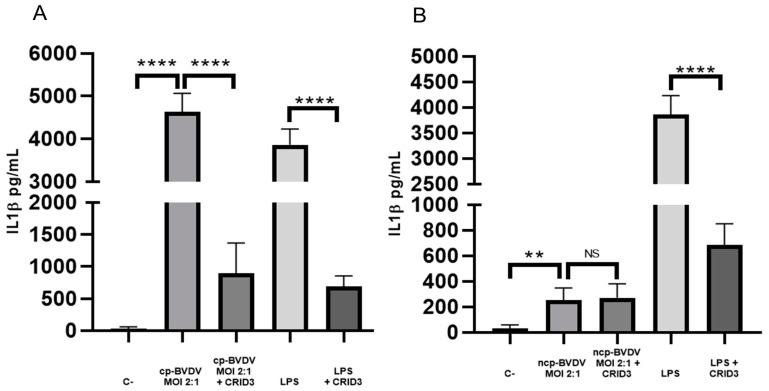
IL-1β secretion is dependent on NLRP3 with the NADL cp-BVDV strain. ELISA of IL-1β from the supernatant of macrophages infected with NADL cp-BVDV strain 24 h post-infection (MOI 2:1) (**A**) and NY-1 ncp-BVDV strain 24 h post-infection (MOI 2:1), (**B**). Negative control (C-): RPMI 10% FBS, treatment with the inhibitor CRID3 (50 μM, 2 h pre-incubation); LPS: supernatant of macrophages stimulated with LPS (300 ng/μL). Statistical differences are given as means ± SD values of three independent assays with two internal replicas each (**** *p* < 0.0001; ** *p* < 0.01). There were no significant differences in the case of the NY-1 ncp-BVDV strain (*p* > 0.05), NS: not significant *p >* 0.05.

**Figure 4 viruses-15-01494-f004:**
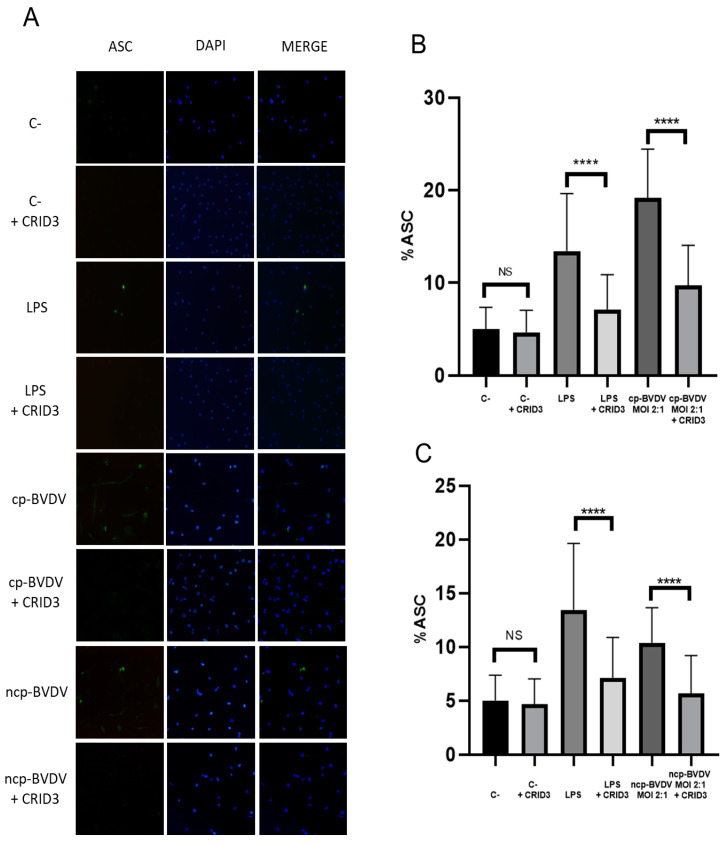
The protein ASC is recruited by NLRP3 during infection with BVDV. (**A**). Representative assay of ASC speck formation using immunofluorescence in macrophages without BVDV (c-), with LPS (300 ng/µL), treated with CRID3 (50 µM, 2 h de pre-incubation), infected with NADL cp-BVDV strain MOI 2:1 and infected with NY-1 ncp-BVDV strain MOI 2:1 (20× objective). (**B**,**C**). Percentages of ASC based on the formation of specks in macrophages without BVDV (c-), with LPS (300 ng/µL), treated with CRID3 (50 µM, 2 h de pre-incubation), infected with NADL cp-BVDV strain MOI 2:1 and infected with NY-1 ncp-BVDV strain MOI 2:1. One-way ANOVA showed significant variation between treatment cells with CRID3- and BVDV-infected cells; LPS: LPS+CRID3 (**** *p* < 0.0001; NS: not significant *p >* 0.05).

**Figure 5 viruses-15-01494-f005:**
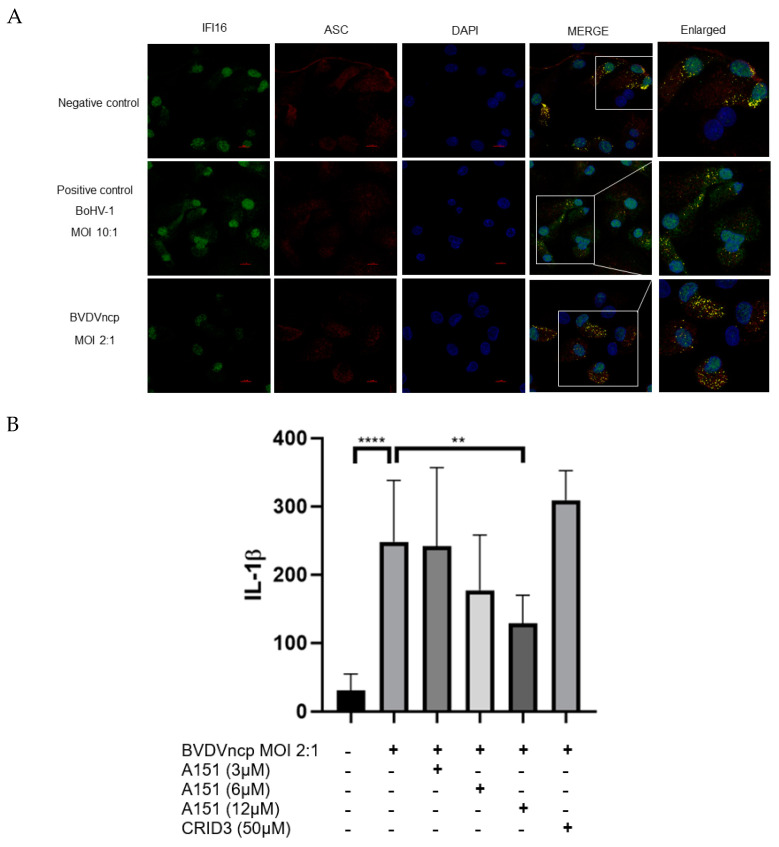
The relationship between IFI16 inflammasome and IL-1β secretion in NY-1 ncp-BVDV infection. (**A**). Colocalization of IFI16 (green) and ASC (red), observed with confocal microscopy in bovine macrophages infected with BoHV-1 MOI 10:1 (positive control) and NY-1 ncp-BVDV strain MOI 2:1 at 24 h.p.i. Negative control: bovine macrophages cultured with RPMI supplemented with 10% FBS. The white boxes indicate the enlarged panels in the last column. (**B**). ELISA of IL-1β from the supernatants of macrophages infected with NY-1 ncp-BVDV strain 24 h post-infection (MOI 2:1) and 2 h pre-incubation with ODN TTAGGG (A151) inhibitor at different concentrations (3 µM, 6 µM and 12 µM). (**** *p* < 0.0001; ** *p* < 0.01).

**Figure 6 viruses-15-01494-f006:**
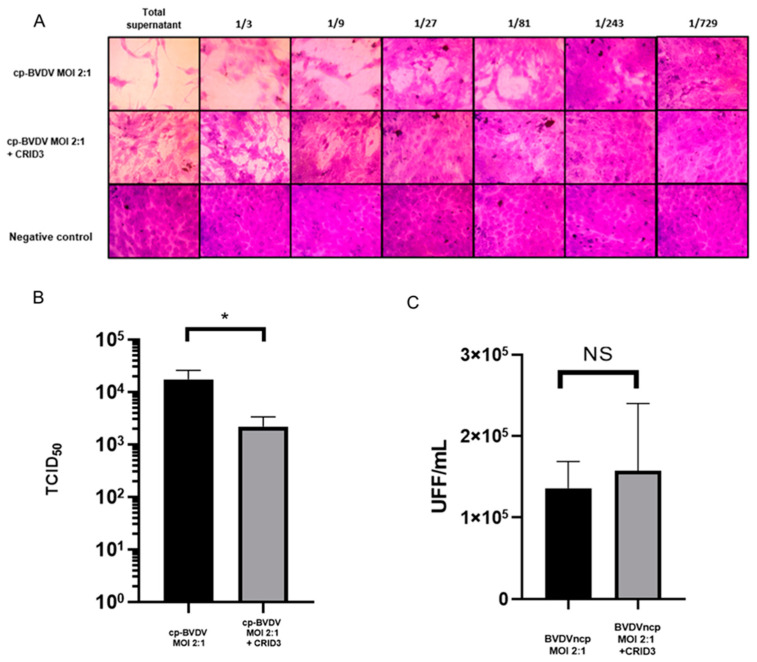
Effect of the CRID3 inhibitor on the viral replication of BVDV. (**A**). Reed and Müench determination in MDBK cells from the infection assays of bovine macrophages infected with NADL cp-BVDV strain (MOI 2:1; 24 h post-infection) without inhibitor and with pre-incubation (2 h) of the inhibitor CRID3 (50 μM); representative pictures of the three independent trials. (**B**). Viral titer of TCID_50_ from the supernatants from the infection assays of bovine macrophages infected with NADL cp-BVDV. (**C**). Viral titers, in FFUs (fluorescent focus-forming units)/mL, of the supernatants from the infection assays of bovine macrophages infected with NY-1 ncp-BVDV. Graphs are representative of the three independent trials. Significant differences according to Student’s *t*-test are indicated (* *p* < 0.05, NS: not significant *p >* 0.05).

**Figure 7 viruses-15-01494-f007:**
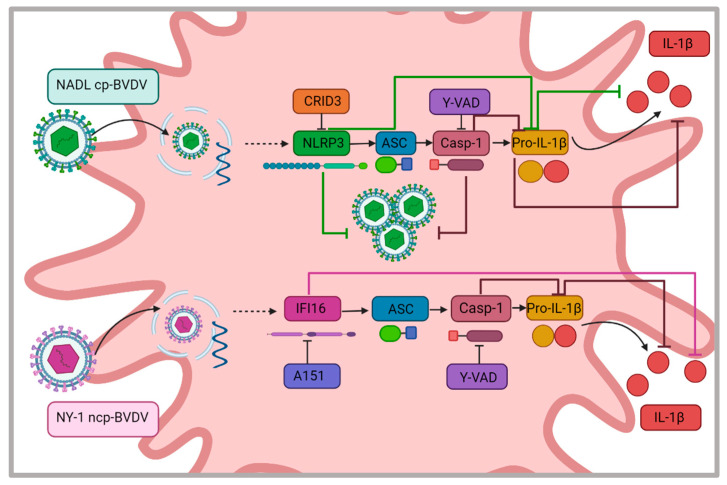
NLRP3 is involved in viral replication of NADL cp-BVDV, and IFI16 is related to secretion of IL-1β with NY-1 ncp BVDV. NADL cp-BVDV promotes the secretion of IL-1β, dependent on inflammasome NLRP3. Inhibition of inflammasome components with CRID3 and Y-VAD decreases cytokine secretion and the viral titer. On the other hand, NY-1 ncp-BVDV induces the secretion of IL-1β through IFI16.

## Data Availability

The datasets generated in this study are available upon request from the corresponding authors.

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
