# Peer review of "NLRP3 Inflammasome Involved with Viral Replication in Cytopathic NADL BVDV Infection and IFI16 Inflammasome Connected with IL-1β Release in Non-Cytopathic NY-1 BVDV Infection in Bovine Macrophages"

_viruses, 2023, doi:10.3390/v15071494_

Round 1
Reviewer 1 Report
General comments: This is a paper that attempts to extend previous work on inflammasomes done with cytopathic BVDV to noncytopathic. The paper suffers from a fatal design flaw. The authors could have used a pair of viruses with same genetic background (TGAC & TGAN; 296c & 296nc or any BVDV pair from MD animal). Instead, they chose two completely unrelated viruses with low virulence. To make any conclusions using NY-1 as the type specific virus is poor design and misleading. This virus has low pathogenesis and rarely causes a febrile response in animals. The statement that NCP induce low inflammatory cytokine levels is also misleading as the virulent NCP BVDV such as 1373 and 890 induce high levels of cytokines (see reference 35 in the paper). The statement that is made near the end of the discussion “However, further studies are needed with this biotype and other high- and low-virulence non-cytopathic strains of BVDV in order to reach a better understanding of the mechanisms of virulence that could be participating [34, 35]” is spot on and should have been considered in the design since reference 35 was published 7 years ago. I believe the previous paper established a good experimental model but the selection of NY-1 as the prototype ncp and any conclusions about NCP based on NY-1 are relatively meaningless. Also, several important references were not included and reference 35 should be more than a closing sentence. Biological significance of the results in this paper are of questionable importance.
Palomares, R.A., Brock, K.V., Walz, P.H., 2014. Differential expression of pro-inflammatory and anti-inflammatory cytokines during experimental infection with low or high virulence bovine viral diarrhea virus in beef calves. Vet Immunol Immunopath 157, 149–154. https://doi.org/10.1016/j.vetimm.2013.12.002
Fan, W., Wang, Y., Jiang, S., Li, Y., Yao, X., Wang, M., Zhao, J., Sun, X., Jiang, X., Zhong, L., Han, Y., Song, H., Xu, Y., 2022. Identification of key proteins of cytopathic biotype bovine viral diarrhoea virus involved in activating NF-κB pathway in BVDV-induced inflammatory response. Virulence 13, 1884–1899. https://doi.org/10.1080/21505594.2022.2135724
Mirosław, P., Rola-Łuszczak, M., Kuźmak, J., Polak, M.P., 2022. Transcriptomic Analysis of MDBK Cells Infected with Cytopathic and Non-Cytopathic Strains of Bovine Viral Diarrhea Virus (BVDV). Viruses 14, 1276. https://doi.org/10.3390/v14061276
Specific comments: The abstract contains a number of abbreviations which are not defined. CRID3 is a acronym that few readers would recognize
The paper is fairly well written grammatical and has just a few spelling mistakes (cytophatic; efect).
Author Response
Dear Editor and Reviewers
We would like to thank you for your observations and recommendations on our manuscript ID 2403530 "please see the attachment"

Reviewer 2 Report
The manuscript brings a new light about viral replication and potential role of NLRP3, caspase 1 and IL-1 beta for BVD infection, mainly non cytopathic BVDV strain.
Considering that non cytophatic strain is more relevant to BVDV pathogenesis and maintain the disease into the herd, researchs on terapies focusing on immune response are benefitials.
I just only to suggest to authors about certification of the fetal bovine serum (FBS) for BVDV. Was it free against BVDV antibodies and BVDV also? If was free of virus and antibodies I suggest to add this information in the Discussion topic. The presence of antibodies would neutralize the action of BVDV into the MDBK cell. Also the presence of BVDV in FSB would competition with non cytopathic BVDV (NY strain) used for the infection experiment.
Author Response

(The authors gave the same response as above.)

Reviewer 3 Report
In this paper, the authors compared the effect of NLRP3 on IL-1β secretion in cp-BVDV and ncp-BVDV infected MDBK cells. And they found that NLRP3 played a vital role in regulating the IL-1β secretion and viral replication of cp-BVDV. However, the IL-1β secretion was not related with NLRP3 in ncp-BVDV infected cells. The datas in this paper is not novel. An onlined paper “Identification of key proteins of cytopathic biotype bovine viral diarrhoea virus involved in activating NF-κB pathway in BVDV-induced inflammatory response” already reported the same result about cp-BVDV. As far as I know, HS (Horse Serum) rather than FBS (Fetal Bovine Serum) is used in researches related with BVDV. In the part of 2.3, auther expressed that both of cp-BVDV and ncp-BVDV propagated on MDBK cells. However, why the viral titration of ncp-BVDV was carried out on BT cells?
There are many grammatical mistakes and irregular expressions in this paper. Extensive editing of English language required. For example, “D-MEM” should be revised as “DMEM”,“TCID50%” should be revised as “TCID50”. Besides, commas and periods missed in many sentences.
Author Response

(The authors gave the same response as above.)

Round 2
Reviewer 1 Report
General Comment
Wherever the term NCP or CP biotype is used throughout the manuscript- biotype needs to be removed and NADL CP strain or NY-1 NCP strain should replace it.
Specific Comments
Here is my suggested title.
NLRP3 Inflammasome is involved with Viral Replication with Cytopathic NADL BVDV Infection and IFI16 Inflammasome is connected with IL-1β Release with Non-Cytopathic NY-1 BVDV Infection in Bovine Macrophages.”
The abstract also needs to be edited and refer to the specific strain- here are a couple of suggested edits. Also make use you either use 1 or 2 significant digits- be consistent
The secretion of the IL-1β in bovine macrophages infected with the Bovine Viral Diarrhea Virus (BVDV) cytopathic strain NADL is caspase 1 dependent. In the present study, we found that in macrophages infected with cytopathic strain NADL, the NLRP3 inflammasome participated in the maturation of IL-1β as levels decreased from 4629.3 pg/ml to 890.09 pg/ml after treatment with cytokine release inhibitory drug 3 (CRID3).
In the case of the non-cytopathic BVDV strain NY-1, IL-1β secretion was not affected by NLRP3 but it may be connected with IFI16 inflammasome. There was a dose response with different concentrations of ODN TTAGGG (A151) inhibitor-Why is this important? In addition, using confocal microscopy, there was colocalization of IFI16 with ASC in the bovine macrophages infected with ncp-BVDV NY-1 strain,
At this point you lost me because it is not clear to me how a qualitative microscopy finding relates to the quantitative measurements that followed
this is related to caspase 1 activation, as evidenced by a decrease in IL-1β from 252.9 pg/mL to 63.5 pg/mL when caspase 1 was inhibited with Y-VAD; this is a novel finding in a ncp-BVDV strain.
On rereading the manuscript needs substantial editing of English
Author Response
Thank you for your observations and recommendations on our manuscript ID 2403530. Please see the attachment.

Reviewer 3 Report
no suggestion.
no comments.
Author Response
Thank you very much for their constructive comments and valuable recommendations, we send the manuscript with grammar review.